# Environmental Determinants of Healthcare Demand: An Explainable ML Approach for Lisbon's Air Quality-Health Nexus

Helder Relvas[*1], Behrouz Nemati[1], Sina Ataee[1], Farzaneh Abedian Aval[1], Vânia Martins[2], Bárbara T. Silva[1], Diogo Lopes[1], and Ana Isabel Miranda[1]

[1]CESAM Department of Environment and Planning, University of Aveiro, 3810-193 Aveiro, Portugal.
[2]Centro de Ciências e Tecnologias Nucleares (C2TN) Department of Nuclear Sciences and Engineering (DECN), Instituto Superior Técnico, Universidade de Lisboa, 2695-066 Bobadela-LRS, Portugal.
helder.relvas@ua.pt

## Abstract

This study applies machine learning (ML) to predict hospital admissions influenced by air pollution and meteorological conditions in Lisbon (Portugal), focusing on Hospital de Santa Maria. Four models, Artificial Neural Networks (ANNs), Random Forest, extreme gradient boosting (XGBoost), and Histogram-Based Gradient Boosting Regressor (HGBR), were trained using air quality (PM2.5, PM10, NO2) and weather variables (temperature, humidity, pressure, wind). HGBR achieved the best performance (Tuning R2: 0.722, Testing R2: 0.521). SHapley Additive exPlanations (SHAP)analysis also showed temperature, particulate matter, and NO2 as key factors. The results highlight that combining gradient boosting with explainable AI provides a reliable, data-driven framework for forecasting hospital demand under changing environmental conditions.

## 1 Introduction

Air pollution is a major environmental factor affecting human health through multiple pathways. It results from both anthropogenic and natural sources, including transportation, industry, energy production, household heating, and natural events such as dust storms or forest fires. Key pollutants include particulate matter (PM10, PM2.5), nitrogen oxides (NOx), sulfur dioxide (SO2), and ozone (O3), which can remain in the atmosphere and enter the human body through inhalation [1,2]. Continuous exposure may induce short-term effects, such as coughing and irritation, and long-term impacts, including respiratory illnesses and cardiovascular disease (CVD) [3,4]. Increases in pollutant levels are often linked to higher hospital admissions (HA), particularly for respiratory conditions [5,6]. Short-term spikes in PM or NOx frequently lead to emergency visits [7,8], whereas chronic exposure increases long-term healthcare demand. Therefore, predicting HA related to air pollution is crucial for assessing health risks and improving public health planning [9,10].

Traditional statistical models, mainly regression-based, are limited in capturing complex nonlinear relationships. Machine learning (ML) techniques, such as artificial neural networks (ANNs) [11], random forest [12], and extreme gradient boosting (XGBoost), [13] can better model the complex interactions between pollutants, weather, and population factors [11,14]. Meteorological variables also play a crucial role in pollutant dispersion and can indirectly affect population vulnerability and health outcomes[15–17].

This study aims to predict hospital admissions in the Lisbon Metropolitan Area (Portugal), focusing on Hospital de Santa Maria. Four ML models are applied to analyze and predict hospital demand, integrating air quality indicators with meteorological variables to identify the key environmental factors influencing admission patterns. By combining these datasets, the study seeks to accurately capture how fluctuations in air pollution and weather conditions affect healthcare demand. The results are expected to provide valuable insights into the relationship between environmental conditions and hospital admissions, offering a predictive framework to help authorities anticipate patient surges and optimize resources.

## 2 Methodology

Hospital admissions were predicted using a data-driven ML approach implemented in Jupyter Notebook. Four ML algorithms were applied, Histogram-Based Gradient Boosting Regressor (HGBR), XG-Boosting, Random Forest, and ANN to evaluate interactions between air pollution and meteorological variables.

The dataset integrated three primary sources covering a 12-year period: (i) Air pollution data (PM2.5, PM10, and NO2) obtained from the QUALAR network; (ii) Hospital admissions data (daily records) provided by Serviço Nacional de Saúde (SNS); and (iii) Meteorological data (2m temperature, 2m dewpoint temperature, surface pressure, 10m wind u and v components) were obtained from ERA5-Land

---
*Corresponding Author.

dataset, extracted for the nearest location of Entrecampos Station, in Lisbon. From this dataset, derived variables (wind speed and direction, relative humidity) were acquired.

# 3   Results

The HGBR model showed the best balance between accuracy and stability among the tested models. The mean coefficient of determination during training was (R2 = 0.521), indicating that the model learned the underlying relationships without significant overfitting. During the tuning phase, HGBR achieved (R2= 0.722), (MAE = 8.26), and (RMSE = 10.57), showing a good trade-off between precision and model simplicity. In the testing phase, the model maintained (R2= 0.521) with controlled error levels, confirming consistent behavior. Parity plots (Figure 1) show predictions closely aligned with observations along the 1:1 line.

The SHAP feature importance analysis (Figure 2) revealed that temperature variables, particularly mean and maximum temperature with seven-day rolling averages, exerted the strongest influence on hospital admissions. This relationship likely reflects the physiological stress of temperature fluctuations on vulnerable groups such as the elderly and those with cardiovascular conditions. PM10 and PM2.5 also ranked among the most impactful features, consistent with their well-known role in triggering respiratory and cardiac events. NO2, representing traffic-related pollution, further emphasized the contribution of urban emission sources to hospital load. Surface pressure and relative humidity showed indirect but consistent effects, probably linked to their control over pollutant dispersion and atmospheric stability.

When comparing model behavior, tree-based algorithms such as HGBR and XGBoost captured complex, nonlinear interactions between environmental variables more effectively than the ANN. This advantage arises from their additive and hierarchical structure, which naturally models delayed or cumulative effects (lags and rolling windows). In contrast, ANN required more extensive tuning and was more sensitive to data size and parameter configuration. Overall, the findings indicate that combining gradient-boosting approaches with interpretable tools such as SHAP provides a reliable, transparent, and data-driven framework for analyzing and predicting hospital admissions under variable air pollution and meteorological conditions. The results highlight temperature fluctuations, fine particulate matter, and NO2 as key features.

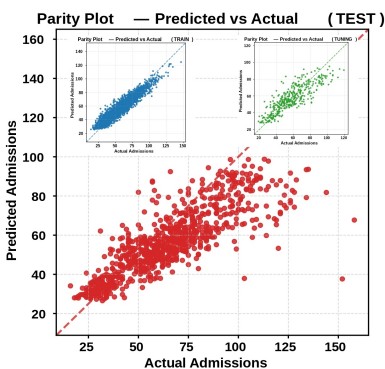

**Figure 1.** Parity plots of predicted versus actual hospital admissions for the highest-performing model (HGBR) across training, tuning, and testing datasets.

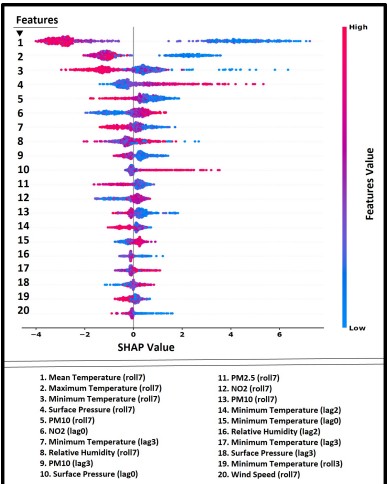

**Figure 2.** SHAP feature importance of the top 20 most influential variables derived from the HGBR.

# 4   Conclusion

This study demonstrates the effectiveness of machine learning in predicting hospital admissions under varying air pollution and meteorological conditions. Among the tested algorithms, the HGBR model showed the best predictive accuracy and generalization, supported by SHAP analysis that identified temperature, particulate matter, and NO2 as key contributors. These findings suggest that integrating gradient-boosting methods with explainable AI can support early-warning systems and improve hospital preparedness through data-driven insights into environmental health risks. Future work will explore deep learning architectures, such as LSTM, to assess whether they can better capture long-term temporal dependencies in multivariate environmental time series, particularly when larger datasets become available.

## Acknowledgments

The authors would like to acknowledge the support of CESAM (UIDP/50017/2020 + UIDB/50017/2020 + LA/P/0094/2020) and C2TN (UIDB/04349/2020). Thanks are due to the ODESSA Project (https://doi.org/10.54499/2024.07293.IACDC). Thanks are due to FCT/MCTES for the contracts granted to Helder Relvas (https://doi.org/10.54499/2021.00185. CEECIND/ CP1659/ CT0026).

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
