# OpenReview forum: "Environmental Determinants of Healthcare Demand: An Explainable ML Approach for Lisbon’s Air Quality-Health Nexus"
_NLDL.org/2026/Abstracts_Track — NLDL 2026 Abstracts_

### Official Review · Reviewer_GuXk · 2025-10-24

**Soundness:** 3
**Correctness:** 3
**Rating:** 4
**Confidence:** 3

**Summary:**

This work applies 4 different machine learning (ML) to predict hospital admissions influenced by air pollution and meteorological conditions in Lisbon focus on using. The models include neural network, gradient boosting regressors and random forests. The feature importance of the model is then examined by through the use of SHAP.
The dataset is generated from three different dataset combined which is related to climate and healthcare data. The different metrics used to evaluate the performance of the model include the R2, MAE and RMSE. Analysis from SHAP showed temperature, fine particulate matter and NO2 were key features. Parity plots were also used to compare the predictions of the model compared to the true predictions.

**Strengths:**

The paper is presented well, it is clear and well-written overall.
The paper focuses on an important area, which is in the intersection of climate and healthcare.
The paper is well motivated, talking about the benefits of knowing how different pollutants can affect the hospital admissions.
The paper looks at several different models rather than a single model.
The plots shown (parity plot and the feature importance plots) are clear and show a lot of useful information.

**Weaknesses:**

The preliminary model performance does not seem too good since it seems to have  a performance of R2 = 0.5
Experiments were done with 4 models however it seems all the information we see is related to a single model, it may be beneficial to see a table describing the performance related to a single model.

---

### Official Review · Reviewer_NN78 · 2025-10-31

**Soundness:** 3
**Correctness:** 3
**Rating:** 4
**Confidence:** 4

**Summary:**

This study investigated the relationship between air pollution and hospital admissions, focusing on how environmental factors influence patient surges. Four machine learning models namely Artificial Neural Networks (ANNs), Random Forest, Extreme Gradient Boosting (XGBoost), and Histogram-Based Gradient Boosting Regressor (HGBR) were trained using air quality indicators (PM2.5, PM10, NO₂) and meteorological variables (temperature, humidity, pressure, wind) to predict hospital admissions. The models were compared to evaluate their predictive performance under varying pollution and weather conditions. Results showed that HGBR achieved the highest predictive accuracy, suggesting that such models could help authorities anticipate patient surges and optimize healthcare resource allocation.

**Strengths:**

This works provides a comprehensive comparison of four machine learning models (ANN, Random Forest, XGBoost, HGBR), offering insights into which methods perform best under varying air pollution and meteorological conditions. By incorporating both air quality and meteorological factors, the study captures the combined effects of environmental conditions on hospital admissions. The results are practically relevant, as accurate predictions could help authorities anticipate patient surges and optimize healthcare resources. The feature importance analysis adds interpretability by showing which variables most strongly influence admissions. In addition, the problem is well-motivated, with a short overview of related work and the background of the proposed solution.

**Weaknesses:**

The study lacks details regarding the experimental setup. Information on the dataset size, including the division into training, tuning, and testing subsets, is not provided. Additionally, the description of how the experiments were conducted and which hyperparameters were used is unclear. In the results section, only the performance of HGBR is reported, while the outcomes of the other three models are omitted. Including a comparative table of all four methods would provide a more comprehensive view of the model performance.

---

### Official Review · Reviewer_YUyz · 2025-11-03

**Soundness:** 2
**Correctness:** 3
**Rating:** 2
**Confidence:** 3

**Summary:**

The work involves developing 4 regression ML models relating hospital admissions to environmental variables. Further, it also carries out SHAP analysis to infer the importance of environmental factors affecting the cases. The authors indicate that the HGBR model performs the best.

**Strengths:**

The work is a focused experimental work, using established ML models, and SHAP analysis for explainability.
It is clearly written and is easy to understand.
The results are reasonably good suggesting a viable relationship between the environmental factors and hospital admissions.

**Weaknesses:**

1) Considering general hospital admissions may lead to a coarse level analysis.
It is important to consider specific conditions related to environmental factors (e.g. thoracic issues, allergies, extreme heat and cold related conditions etc.), Without this the effectiveness of the study is limited

2) The SHAP analysis can be treated differently for the environmental factors and the meteorological variables. Presently, the latter are dominating the former,

3) While only the results of the best model are provided, perhaps a more fine-grained comparison of all the models can be given. Sometimes there may be pros and cons between the models, in terms of the fit of the data in specific ranges or conditions.

4) Some intuition about why HGBR performs better, can be discussed, which would provide a sound basis of why the performance is good, and may be useful for the readers to extend it for other applications.

---

### Decision · Program_Chairs · 2025-11-05

**Decision:**

Accept

**Comment:**

The reviewers found the abstract borderline, yet the PCs believe it will be of interest to the community and should have the opportunity be presented.